# Best Supportive Care for Patients with Chronic Lymphocytic Leukemia: Relevance of Cancer Screening and Immunizations

**DOI:** 10.3390/cancers17132093

**Published:** 2025-06-23

**Authors:** Vanthana Bharathi, Alessandra Ferrajoli

**Affiliations:** Department of Leukemia, MD Anderson Cancer Center, Houston, TX 77030, USA; vbharathi@mdanderson.org

**Keywords:** CLL, vaccine, cancer screening, health maintenance

## Abstract

**Simple Summary:**

Chronic lymphocytic leukemia (CLL) is a type of blood cancer that affects B cells, and it is typically seen in older adults and often leads to weakened immune system. With newer treatments helping people live longer, it is important to shift the focus on health maintenance care. This review explores how supportive care strategies such as vaccines, cancer screenings, and healthy lifestyle changes can help people with CLL stay healthier and avoid infections and detect cancers at earlier stages. The article emphasizes the need for timely vaccinations, cancer screenings, and tailored preventive strategies. These steps can improve quality of life, decrease morbidity, and reduce the risk of hospitalizations. Our goal is to provide healthcare providers with practical recommendations to enhance long term care for people living with CLL.

**Abstract:**

Chronic lymphocytic leukemia (CLL) is the most prevalent leukemia in the Western world, predominantly affecting older individuals. Advances in targeted therapies have extended survival, shifting the clinical focus toward the management of chronic health challenges. Patients with CLL often experience immune dysfunction due to both the disease and its treatments, resulting in an increased susceptibility to infections and second primary malignancies. This review outlines evidence-based strategies for preventive care in CLL, including age-appropriate cancer screenings, routine immunizations, and lifestyle modifications. By emphasizing proactive health maintenance, this article aims to support clinicians in delivering comprehensive, long-term care for patients with CLL.

## 1. Introduction

Chronic lymphocytic leukemia (CLL) is the most prevalent leukemia in the Western world; it affects over 200,000 people in the United States and is more common among the older population [1]. Over the past decade, targeted therapies, such as inhibitors of Bruton tyrosine kinase (BTK) and B-cell lymphoma 2 anti-apoptotic protein-2 (Bcl-2), have revolutionized the management of CLL. These targeted therapies have substantially improved survival rates and shifted the disease course towards a chronic, manageable illness for most patients. Despite these therapeutic advancements, clinicians caring for patients with CLL still face several challenges [2]. Long-term survivors of CLL are at risk for health complications such as an increased rate of infections, a suboptimal response to immunizations, cardiovascular toxicities associated with treatment, and an increased rate of other cancers [3]. Consequently, healthcare providers should prioritize providing a comprehensive survivorship care plan to address the evolving needs of this growing patient population. The best supportive and preventive care encompasses a comprehensive approach to optimizing outcomes in patients with CLL. In this narrative review, we summarize recommended cancer screenings and vaccination strategies for patients with CLL based on current professional society guidelines (ASCO, NCCN, CDC, and IDSA), and expert consensus. While the discussion primarily reflects U.S. clinical practice, we acknowledge that vaccine availability, treatment access, and reimbursement policies vary globally, and recommendations should be adapted based on regional healthcare systems and resources.

## 2. Immune Dysfunction in CLL: A Multifaceted Phenomenon

Immune dysfunction is a hallmark of CLL, and, as the disease progresses, patients experience increasing levels of immunosuppression. This dysfunction affects both the innate and adaptive arms of the immune system. Innate immune abnormalities may include low complement levels and functional complement deficiencies, while adaptive immune impairment is characterized by T-cell dysfunction and hypogammaglobulinemia [4]. These immune alterations impair host defense mechanisms, increasing the susceptibility to infections and other cancers.

### 2.1. Treatment-Related Immune Dysfunction

For patients that have a progression of disease and require treatment, immune dysfunction becomes more pronounced since both disease- and treatment-related immune dysfunction develop. Historically, chemotherapy, such as the fludarabine, cyclophosphamide, and rituximab (FCR) regimen, and the bendamustine and rituximab (BR) regimen, induced profound immune suppression. Fludarabine causes T-cell depletion, while rituximab leads to B-cell depletion and the subsequent hypogammaglobulinemia [4]. Similarly, bendamustine causes lymphopenia and CD4 T-cell count suppression, and, since it is given in combination with rituximab, this regimen is accompanied by B-cell depletion and hypogammaglobulinemia.

More recently, oral targeted therapies which have become standard in CLL can also impact the immune system [5], but their effects on immune function are less well-understood. BTK inhibitors, by blocking chronic active B-cell receptor (BCR) signaling in malignant B cells, alter neo-plastic cell migration and apoptosis [6]. However, BTK inhibitors also modulate non-malignant cells and contribute to immunosuppression. Since BTK is expressed in monocytes/macrophages, dendritic cells, neutrophils, and mast cells, its inhibition alters the immune environment, reducing inflammation and resulting in an overall suppressed immune response [7]. Similarly, BCL-2 inhibitors have been shown to cause cytotoxic T-lymphocyte depletion and can be associated with neutropenia, further contributing to immunosuppression [8]. Phosphoinositide 3-kinase (PI3K) inhibitors induce secondary immune suppression in CLL primarily through their effects on T cells and NK cells. Idelalisib, a selective inhibitor of the PI3Kδ isoform, has been shown to reduce the expression of inhibitory checkpoint molecules on T cells, decreasing T-cell-mediated cytotoxicity, granzyme B secretion, and cytokine production, and diminishing the proliferation and cytotoxicity of NK cells [9].

### 2.2. Clinical Implications: Infection Risk and Secondary Malignancies

As a result of the cumulative immune impairment, patients with CLL face an increased risk of infections compared to the general population. Furthermore, hypogammaglobulinemia has long been recognized as a risk factor for the development of infections. A large retrospective study found that patients with CLL and secondary immunodeficiency were significantly more likely to experience infections (70.1% vs. 30.4%) and severe bacterial infections (39.8% vs. 9.2%) compared to those without secondary immunodeficiency [10]. This cohort also had a higher use of antimicrobials, greater healthcare resource utilization, and shorter overall survival (12.3 vs. 16.9 months).

Additionally, the impaired immune surveillance in CLL contributes to a heightened risk of second primary malignancies (SPMs). Several studies have documented that patients with CLL have an elevated risk of developing SPMs. A recent study reported that second malignancies were identified in 36% of CLL patients. Compared to the general population, long-term survivors of CLL exhibited a significantly increased risk of developing other malignancies, with the risk being more pronounced in males and patients under 60 years of age [11].

## 3. Cancer Screening in Patients with CLL

With patients with CLL living longer, SPMs have become increasingly relevant and better studied. Several mechanisms have been attributed to the association between CLL and second cancers, including disease-related immune dysregulation, treatment-related immunosuppression, and cumulative chemotherapy exposure. An analysis by Falchi et al. investigated the incidence and prognostic impact of other cancers in long-term survivors of CLL. The study included 797 patients who survived more than 10 years post-CLL diagnosis. The cumulative frequency of other cancers was 36%, with non-melanoma skin cancer being the most common, followed by prostate, breast, melanoma, lung cancers, and secondary leukemia. The standardized incidence ratio (SIR) for all other cancers was 1.2, indicating a 20% increased risk compared to the general population. The study found that advanced age, male gender, and lower platelet counts were independent predictors of developing other cancers. Importantly, the presence of another cancer was associated with a significantly shorter survival (16.2 months) compared to those without another cancer (22.9 years) [11]. Zheng et al. (2019) [12] conducted a study using the Swedish Family Cancer Database to assess the risk of second primary cancers in patients with CLL, acute lymphoblastic leukemia, and hairy cell leukemia. The study found significant relative risks (RRs) for CLL for 20 different second primary cancers. Notably, the risks were particularly high for skin squamous cell cancer (RR 24.58 for in situ and 7.63 for invasive), Merkel cell carcinoma (RR 14.36), Hodgkin lymphoma (RR 7.16), and Kaposi sarcoma (RR 6.76) [12]. Given this growing body of evidence, cancer screening is critically important for patients with CLL. Routine surveillance and tailored interventions—including regular skin exams, lung cancer screening in high-risk individuals, and age-appropriate screening for other common malignancies—should be integrated into survivorship care plans (Table 1) to mitigate long-term morbidity and mortality.

### 3.1. Skin Cancer Screening

Patients with CLL face a heightened risk of skin cancers, particularly cutaneous squamous cell carcinoma (cSCC) and melanoma. Environmental and phenotypic factors further amplify this risk. Ultraviolet (UV) radiation is a well-established carcinogen, and regions with a high UV index—such as Australia, the southern part of the United States, and Mediterranean Europe—pose a particularly elevated risk for skin cancer in patients with CLL. The typical demographic profile of patients with CLL includes individuals over the age of 70, with a 2:1 male-to-female ratio and a predominance of white individuals—approximately 90% of CLL cases in Western populations occur in non-Hispanic whites. Individuals with lighter skin tones (Fitzpatrick skin types I and II) are more susceptible to UV-induced DNA damage due to the reduced melanin protection. Both UVA and UVB radiation contribute to carcinogenesis: UVB (290–320 nm) is primarily responsible for direct DNA damage and sunburns, while UVA (320–400 nm) penetrates deeper into the skin and causes oxidative stress and indirect DNA damage, both of which can lead to skin cancers. Chronic cumulative exposure to UVA and intermittent intense UVB exposure (e.g., sunburns) are strongly associated with squamous cell carcinoma and melanoma, respectively. In this context, older white males with prolonged cumulative sun exposure represent a high-risk subgroup. Studies have shown that, even in regions with a relatively low UV exposure, such as Manitoba, Canada, the incidence of skin cancer in patients with CLL remains substantial, underscoring the synergistic role of immune dysfunction and UV sensitivity [13]. Therefore, patient education and dermatologic surveillance should be adapted based on both individual and geographic risk factors.

The risk of melanoma in patients with CLL is markedly elevated, with a standardized incidence ratio (SIR) of 7.74 compared to the general population. This increased risk is consistent across different age groups and persists for many years following the diagnosis of CLL [14]. A retrospective case–control study by Jobson et al. (2022) [15] found that CLL was associated with significantly worse melanoma-specific mortality (hazard ratio [HR] 2.46, 95% confidence interval [CI] 1.27–4.74), and recurrence (HR 3.44, 95% CI 1.79–6.63) [15]. Fisher et al. (2022) [16] reported that melanoma-specific mortality was higher in patients with a prior diagnosis of CLL, with 17.9% of patients dying from melanoma compared to 5.8% in those diagnosed with melanoma before developing CLL. This study also noted that melanoma recurrence and distant metastases were more common in patients with a prior CLL diagnosis, further underscoring the aggressive nature of melanoma in this population [16].

Non-melanoma skin cancers, SCC, and basal cell carcinoma (BCC) are also more common in patients with CLL. The incidence of SCC is particularly high, with an SIR of 24.58 for in situ SCC and 7.63 for invasive SCC. Interestingly, the ratio of SCC to BCC is reversed in patients with CLL when compared to the general population, with SCC being more prevalent than BCC [12]. As a result, regular skin cancer screening is of crucial relevance in these patients. A survey at our center evaluated sun protection and skin examination practices among 100 patients with CLL. While 70% reported having had a physician skin exam, only 51% had one within the past year, and just 22% had performed a skin self-exam in the past 3 months. Routine use of sunscreen and other sun-protective behaviors was suboptimal in the patients surveyed. These findings reiterate the importance of integrating skin cancer education and routine dermatologic screening into CLL survivorship care.

In a community-based study, 22% of patients who received a full-body skin examination within 6 months of CLL diagnosis were found to have a malignancy, and 21% of the entire cohort developed skin cancer over time. These findings highlight the importance of early dermatologic referral and support current recommendations for a skin examination within 6 months of diagnosis, followed by annual evaluations thereafter [17]. Additionally, patients with CLL should practice strict sun protection, including using broad-spectrum sunscreen, wearing protective clothing, and limiting intense sun exposure, to help reduce their risk of sunburn and reduce the risk of developing new skin cancers.

### 3.2. Breast Cancer Screening

According to the study by Falchi et al., breast cancer was among the most common second cancers observed in both treated and untreated long-term survivors of CLL [11]. For breast cancer screening, the American Society of Clinical Oncology (ASCO) and the National Comprehensive Cancer Network (NCCN) recommend that cancer survivors, including those with CLL, generally follow the same guidelines as the general population, with modifications based on individual risk factors. Women with CLL should undergo an annual mammography starting at age 40 or earlier if they have additional risk factors, such as a family history of breast cancer. Clinical breast examinations are advised every three to four months for the first three years after CLL diagnosis, then every six months for the next two years, and annually thereafter. Additionally, breast MRI should be considered for high-risk individuals, particularly those with a strong family history of breast cancer or known genetic predispositions [18]. A discussion regarding the risks and benefits of continuing screening after age 65 is advisable in patients with CLL and should be based on individual risk factors. Additionally, it is essential that patients share with the clinical personnel their diagnosis of CLL when undergoing a mammogram since abnormal lymph nodes related to CLL may be present in the exam and need to be interpreted based on clinical information.

### 3.3. Cervical Cancer Screening

For cervical cancer, ASCO and NCCN recommend Pap tests every 3 years for women aged 21–29, and either a Pap test and human papillomavirus (HPV) co-testing every 5 years or a Pap test alone every 3 years for immunocompetent women aged 30–65.

The guidelines for cervical cancer screening in immunocompromised patients, including those with CLL, are based on recommendations for individuals with similar immunosuppressive conditions, such as HIV infection and solid organ transplantation. The American Society for Colposcopy and Cervical Pathology (ASCCP) suggests that immunocompromised individuals should undergo screening annually for the first 3 years after diagnosis, then undergo cytology-only screening every 3 years until age 30, and, subsequently, either continue with cytology alone or co-testing every 3 years [19].

### 3.4. Colorectal Cancer Screening

For colorectal cancer, the ACS (American Cancer Society) and NCCN recommend that individuals at higher risk, such as those with a history of cancer or immunocompromised conditions, undergo intensive surveillance. The recommendation includes a colonoscopy every 10 years starting at age 45 for average-risk individuals. However, earlier and more frequent screening may be warranted for those with CLL based on individual risk factors. If a colonoscopy is not feasible, annual fecal immunochemical testing (FIT) or high-sensitivity guaiac-based fecal occult blood testing (gFOBT) can be used, while a flexible sigmoidoscopy and CT colonography every 5 years are also considered acceptable alternatives [20]. A discussion regarding the risks and benefits of continuing screening after age 75 is advisable in patients with CLL and should be based on individual risk factors and patient/physician preferences.

### 3.5. Lung Cancer Screening

In 2023, ACS updated its lung cancer screening guidelines, recommending annual low-dose computed tomography (LDCT) for adults 50 to 80 years old with a 20-pack-a-year smoking history who currently smoke or quit within the past 15 years [21]. This aligns with the guidelines from the United States Preventive Services Task Force (USPSTF), which also emphasize balancing the benefits of early detection with potential harms like false positives, overdiagnosis, and radiation exposure. NCCN supports a similar screening but without an upper age limit, provided the individual is a candidate for curative-intent treatment. Across these major guidelines, an annual LDCT is recommended for high-risk individuals, highlighting the importance of individualized risk assessment and shared decision-making in lung cancer screening.

In contrast, the chest X-ray (CXR) has limited utility for lung cancer screening in asymptomatic individuals. Multiple randomized controlled trials and meta-analyses have not demonstrated a meaningful reduction in lung-cancer-specific mortality with CXR [22]. Even among symptomatic patients, its sensitivity for detecting lung cancer is modest—ranging from 77% to 80%—and a negative result may not rule out the need for further evaluation in high-risk individuals [23].

**Table 1 cancers-17-02093-t001:** Suggested cancer screening recommendations with special considerations for CLL.

Cancer Type	Screening Test	Recommended Age/Frequency	Special Considerations for CLL	References
Skin Cancer	Full-body skin exam	Skin exam annually	Dermatologist exam within 6 months of diagnosis, then annually.	NCCN guidelines, [17]
Breast Cancer	Mammography ± MRI for high-risk	Annually from age 40, earlier if high-risk	Consider MRI for patients with genetic predisposition or family history, consider continuing screening past age 74.	NCCN guidelines, [18]
Cervical Cancer	Pap test ± HPV co-testing	Pap test every 3 years from 21–29, and then pap test or HPV co-testing every 3–5 years	Annually for 3 years, then every 3 years with cytology alone until age 30. After 30, continue cytology every 3 years or co-testing with HPV every 3–5 years.	ASCCP guidelines, [19]
Colorectal Cancer	Colonoscopy, FIT, or gFOBT	Colonoscopy every 10 years from age 45, or earlier if high-risk	Consider continuing screening past age 75.	ACS/NCCN guidelines [20]
Lung Cancer	Low-dose CT scan	Annually from age 50 if ≥20 pack-years smoking	Ensure shared decision-making due to potential risks of overdiagnosis.	ACS guidelines
Prostate Cancer	PSA test ± digital rectal exam	Discuss screening at age 50 (45 if high-risk)	Consider continuing screening past age 69.	ACS guidelines, [24]

Nonetheless, CXR continues to be widely used in low- and middle-income countries, where access to LDCT remains limited. Emerging research is investigating the role of artificial intelligence in improving CXR interpretation in resource-limited settings, though such technologies are not yet part of standard practice.

### 3.6. Prostate Cancer Screening

Prostate cancer is a significant health concern, but the benefits of screening remain debated. Patients with CLL face an increased risk of prostate cancer. In the study by Falchi et al. among long-term survivors of CLL, prostate cancer was the second most common cancer after non-melanoma skin cancer [11], making individualized screening essential.

ACS recommends that asymptomatic men with a 10-year life expectancy discuss screening with their provider. For average-risk men, discussions start at age 50; for higher-risk individuals (African American men, CLL patients, or those with a family history before 65), at age 45; and, for highest-risk men (multiple family diagnoses before 65), at age 40 [24]

USPSTF advises individualized PSA screening for men aged 55 to 69 while discouraging it for those 70 and older due to potential harms [25]. Given the elevated cancer risk in patients with CLL, screening decisions should consider overall health, life expectancy, family history, and personal preference.

## 4. Immunizations in Patients with Chronic Lymphocytic Leukemia

Patients with CLL face an increased risk of infections due to the immunosuppressive nature of the disease and the effects of treatment, including BTK inhibitors and anti-CD20 monoclonal antibodies. Patients are particularly vulnerable to infections caused by encapsulated bacteria such as Streptococcus pneumoniae and Haemophilus influenzae [26]. However, due to widespread pediatric vaccination and resulting herd immunity, infections from Haemophilus influenzae type b (HiB) have become exceedingly rare in most populations, and some countries have removed the HiB vaccine from adult immunization schedules. In contrast, non-typeable Haemophilus influenzae strains—which are not vaccine-preventable—remain more common causes of pneumonia in immunocompromised adults, including those with CLL. Additionally, influenza and herpes zoster are common and often associated with severe complications [27]. Opportunistic fungal infections may also occur, especially in patients receiving chemo-immunotherapy.

Targeted therapies further modulate infection risk. BTK inhibitors are associated with bacterial respiratory infections and opportunistic infections like Pneumocystis jirovecii pneumonia (PJP) and invasive fungal infections, due to BTK inhibition impairing immune cell signaling. PI3K inhibitors confer an even higher risk, including PCP and cytomegalovirus reactivation, as PI3Kδ plays a central role in B-cell function. BCL-2 inhibitors are less immunosuppressive but may cause neutropenia, increasing the risk of bacterial infections. Infection prevention strategies—including vaccination, prophylactic antimicrobials, and close monitoring—are essential in order to mitigate these risks.

Vaccination remains a cornerstone of infection prevention in CLL; however, patients with CLL have inherent immune defects that impair vaccine efficacy and responses to vaccinations are often attenuated, even in untreated patients. This attenuation is further amplified in those treated with anti-CD20 monoclonal antibodies, BTK inhibitors, and BCL-2 inhibitors because of the transient or persistent hypogammaglobulinemia and disrupted B-cell function that are essential for effective vaccine-induced immunity. ASCO emphasizes that both T- and B-cell activation are important for robust vaccine responses and, even if B-cell dysfunction is often persistent, some patients may retain or recover T-cell function, allowing for partial cellular responses, which can still confer some protection [28].

In a study by Douglas et al., only 26% of patients on treatment with BTK inhibitors seroconverted following high-dose influenza vaccination [29]. Similarly, hepatitis B vaccine seroconversion was significantly lower in patients treated with BTKi- (28%) versus those not on therapy (3.8%) [30]. In contrast, the response to the recombinant zoster vaccine was comparable between treatment-naïve patients and those receiving BTKi. This difference likely reflects the underlying immunologic mechanisms: BTK inhibitors primarily impair de novo humoral responses that depend on naïve B-cell activation—such as those triggered by primary hepatitis B vaccination—while recall responses to previously encountered pathogens like varicella zoster virus or measles may remain intact due to the persistence of long-lived plasma cells. This distinction is clinically meaningful, as it implies that patients on BTKi may retain immunity to childhood infections but demonstrate attenuated responses to novel antigens. These insights support the rationale for administering booster vaccines that target recall antigens when feasible [31]. Notably, factors associated with diminished vaccine responses include age >60 years, IgG <400 mg/L, prior CLL therapy, and progressive disease [32].

ASCO and other professional societies recommend the administration of inactivated vaccines in CLL patients at a low threshold, as they are considered safe—even in the setting of immunoglobulin replacement therapy (IVIG). Live vaccines, however, are generally contraindicated due to the underlying immune compromise and should only be considered in consultation with infectious disease specialists. While the optimal vaccination strategy remains unclear, individualized prevention plans—accounting for treatment history, disease status, and immune function—are essential in order to reduce infection-related morbidity and mortality in this vulnerable population (Table 2).

### 4.1. Pneumococcal Vaccine

The Centers for Disease Control and Prevention (CDC) recommends pneumococcal vaccination for all adults aged ≥19 years who are immunocompromised, including those with hematologic malignancies. For healthy individuals, the recommendations include either a single dose of the 20-valent pneumococcal conjugate vaccine (PCV20) or PCV21 or a sequential regimen of the 15-valent conjugate vaccine (PCV15) followed by the 23-valent pneumococcal polysaccharide vaccine (PPSV23), administered at least 8 weeks later. For those receiving PPSV23, a second dose is advised five years after the first, with a final dose after age 65 if not already given. If PCV20 or PCV21 is used, a dose of PPSV23 is not indicated. Regardless of which vaccine is used (PCV20 or PCV21), their pneumococcal vaccinations are complete [33].

Both untreated and treated patients with CLL are at an increased risk of pneumococcal pneumonia, but the risk is further increased in those receiving CLL-directed therapy. In a study by Haggensburg et al., serum IgG antibody levels against nine pneumococcal serotypes (five shared between PCV13 and PPSV23, and four unique to PPSV23) were measured in patients with CLL before and eight weeks after vaccination using a quantitative multiplex immunoassay. The serologic protection rate (SPR), defined as an antibody concentration of ≥1.3 µg/mL for ≥70% of serotypes, was achieved only in 13% of untreated patients and just 3% of those who had received CLL-directed therapy, underscoring the marked impairment of humoral immunity in this population [34].

Hence, it is essential that we administer pneumococcal vaccines early in the disease course—preferably before initiating immunosuppressive therapy—to improve vaccine responsiveness. ASCO and IDSA support vaccination in this population, even though measured immunoresponses may be suboptimal. Individualized vaccination timing and monitoring are essential for reducing the risk of severe pneumococcal infections in patients with CLL.

### 4.2. Influenza Vaccine

CDC recommends annual influenza vaccination for all adults, particularly those immunocompromised, including individuals with hematologic malignancies such as CLL [33]. The annual incidence of influenza in patients with CLL is significantly higher than the general population due to their underlying immune dysfunction. A study by Vilar-Compte et al. highlighted the increased risk of lower respiratory tract infections and mortality in patients with hematologic malignancies, including CLL, following influenza infection [35]. Additionally, in a real-world cohort study by Tey et al., the rate of severe infections, including influenza, was reported to be 65 infections per 100 person-years among patients receiving ibrutinib, idelalisib, or venetoclax for CLL [36].

To mitigate this risk, the American College of Allergy, Asthma & Immunology recommends annual vaccination with the high-dose inactivated influenza vaccine for all patients with CLL regardless of age [37]. While vaccine-induced immunity may be diminished in this population, even partial protection can reduce the risk of severe complications and hospitalization. Immunogenicity studies confirm the reduced vaccine efficacy in CLL. In a study of 30 subjects (17 with CLL and 13 with monoclonal B-cell lymphocytosis [MBL]), only 17.6% of CLL patients achieved seroprotection against influenza, compared to 76.9% in the MBL group (*p* = 0.002), with lower antibody titers also observed in patients with CLL (*p* = 0.01). These findings underscore the importance of timely and individualized influenza vaccination in CLL—preferably before treatment initiation or during disease stability—with a preference for high-dose formulations to maximize the benefit.

### 4.3. COVID-19 Vaccines

CDC recommends COVID-19 vaccination for everyone 6 months and older with the updated 2024–2025 formulations regardless of prior vaccination status.

Patients with CLL are at a high risk for severe COVID-19 outcomes. A multicenter international study by Mato et al. reported that, among 198 CLL patients with symptomatic COVID-19, 90% required hospitalization and 33% died from the infection [38]. Another study by Scarfò et al. found that 79% of 190 CLL patients with confirmed COVID-19 presented with severe disease, requiring oxygen support or ICU admission, with a 36.4% mortality rate in those with severe illness [39].

Hence, for individuals who are moderately or severely immunocompromised—such as patients with CLL— CDC and ASCO advise completing a multi-dose initial series with an mRNA vaccine such as BNT162b2 (Pfizer) or mRNA-1273 (Moderna) and receiving booster doses every six months or earlier based on clinical need. Additional doses may be considered under shared clinical decision-making to maintain protective immunity.

The evidence suggests that multiple doses improve immune responses. In a study by Shen et al., 73.7% of patients with CLL became seropositive after receiving three doses, underscoring the importance of booster doses in this population [40].

### 4.4. Herpes Zoster Vaccine

Herpes zoster infection—commonly presenting as shingles or dermatomal zoster—is a frequent and serious complication in patients with hematologic malignancies, including CLL, with the risk being the highest within the first two years following diagnosis. ASCO recommends the recombinant zoster vaccine (RZV, Shingrix) for all cancer patients, including those with CLL, due to its superior safety and efficacy compared to the live-attenuated zoster vaccine.

The RZV is administered in two doses, typically spaced two to six months apart, though the interval may be shortened to four weeks in cases where rapid protection is needed. Studies have demonstrated that RZV is immunogenic and safe in patients with CLL. Diamantopoulos et al. reported that it significantly boosts IgG levels, even in those actively receiving treatment or heavily pretreated, with an acceptable safety profile [41]. However, Muchtar et al. found that patients with CLL on BTK inhibitors had reduced humoral and cellular responses to the vaccine [42].

### 4.5. Other Non-Live Vaccines

In addition to pneumococcal, influenza, COVID-19, and herpes zoster vaccines, patients with CLL should also receive age-appropriate and risk-based vaccinations, including the hepatitis B and tetanus, diphtheria, and acellular pertussis (Tdap) vaccine. Hepatitis B vaccination is particularly important in patients with CLL due to the potential risk of exposure during medical care and the potential for viral reactivation in patients undergoing immunosuppressive therapy, especially those receiving anti-CD20 monoclonal antibodies or other B-cell depleting agents. In the United States, the CDC recommends universal hepatitis B vaccination for adults aged 19–59 years and risk-based vaccination for those ≥60 years or with immunosuppression, including hematologic malignancies [28].

Live-attenuated vaccines, such as MMR (measles, mumps, rubella), varicella, and intranasal influenza, should generally be avoided in patients with CLL due to the risk of vaccine-associated disease in the setting of impaired cellular immunity. This is particularly critical for patients undergoing active treatment or with profound immune suppression. Exceptions may apply under specific circumstances in long-term survivors or those in remission, and decisions should be made in consultation with infectious disease specialists [28].

Vaccination planning is also essential for patients with CLL who are considering international travel. Depending on the destination, certain vaccines—such as those for hepatitis A, typhoid, yellow fever, or Japanese encephalitis—may be recommended. While some of these vaccines are live and contraindicated, inactivated alternatives or travel-specific risk mitigation strategies may be appropriate. Early pre-travel consultation is strongly advised to allow sufficient time for proper immunization, especially in those requiring multi-dose series [28].

### 4.6. Timing of Vaccination

To optimize the immune response, vaccinations should ideally be administered early in the course of the disease and, when possible, before the initiation of treatment or during periods of disease control if treatment has already started [37]. The goal is to enhance vaccine efficacy and reduce the likelihood of severe infections.

For patients on BTKi therapy, pausing treatment around the time of vaccination is not recommended. The randomized IMPROVE trial, which included 99 patients, found no significant difference in antibody response between those who paused BTKi and those who continued therapy (geometric mean ratio 1.10; 95% CI, 0.57–2.16; *p* = 0.77) [43]. These findings reinforce that BTKi should not be interrupted for vaccination purposes.

For patients receiving repeated courses of anti-CD20 monoclonal antibodies, B-cell reconstitution becomes increasingly impaired with each exposure. In those on continuous anti-CD20 therapy, seasonal influenza vaccination is recommended at least 4 weeks after the most recent dose. For nonseasonal vaccines, administration should ideally occur 2–4 weeks prior to starting anti-CD20 treatment or be deferred until 6–12 months after its completion when possible [28].

Vaccination guidelines for patients undergoing CAR-T cell therapy remain limited due to the sparse data. When feasible, non-live vaccines should be administered before CAR-T therapy or delayed until at least 6–12 months post-treatment. Influenza and COVID-19 vaccines are best given at least 2 weeks prior to lymphodepleting chemotherapy or following the same timing recommended for hematopoietic stem cell transplant (HSCT) recipients (i.e., ≥3 months after CAR-T). No data currently support the safety or timing of live vaccines in this population, and the need for revaccination or specific vaccine schedules remains poorly defined [28].

### 4.7. Adjuncts to Vaccination

For additional protection against COVID-19, monoclonal antibodies such as pemivibart (Pemgarda™) have emerged as a pre-exposure prophylaxis option for immunocompromised individuals, including those with CLL, who are unlikely to mount an adequate immune response to vaccination. Pemivibart received Emergency Use Authorization from the FDA in March 2024 for pre-exposure prophylaxis against COVID-19 in moderately to severely immunocompromised individuals. This approval was based on immunobridging data from the CANOPY trial, which demonstrated that pemivibart produced neutralizing antibody titers consistent with those seen in other monoclonal antibodies with known clinical efficacy [44]. The American Society for Transplantation and Cellular Therapy (ASTCT) recommends considering the administration of monoclonal antibodies toward certain viruses for very high-risk patients who may have an inadequate vaccine response, such as recipients of cellular therapy.

## 5. Special Considerations in CLL Supportive Care

### 5.1. Antimicrobial Prophylaxis

Patients with CLL who experience severe immunosuppression require antimicrobial prophylaxis, as recommended by multiple professional societies. ASCO and the Infectious Diseases Society of America (IDSA) advise prophylaxis for high-risk patients with CLL to prevent serious infections. NCCN guidelines recommend prophylaxis against PJP pneumonia with trimethoprim-sulfamethoxazole (TMP-SMX) or an alternative agent for patients receiving PI3K inhibitors, purine-analog- or bendamustine-based chemo-immunotherapy, and/or alemtuzumab [45]. Additionally, prophylaxis against Varicella zoster virus infections with antiviral agents such as valacyclovir is recommended for patients undergoing similarly high-risk treatments [46]. In contrast, the risk is substantially lower in patients managed with BTKi alone, where the incidence of PJP is generally below 3% and routine prophylaxis is not universally recommended [47].

Hepatitis B reactivation is a well-recognized complication of B-cell-depleting therapies, particularly anti-CD20 monoclonal antibodies. Therefore, hepatitis B virus screening with the surface antigen HBsAg, core antibody, and surface antibody should be performed prior to initiating such treatments. Prophylactic antiviral therapy should be strongly considered in patients with a positive HBV serology to prevent reactivation during and after B-cell-depleting therapy.

### 5.2. IVIG Replacement Therapy in CLL

Intravenous immunoglobulin (IVIG) therapy should be restricted to a carefully selected subset of patients with CLL. Patients with CLL, secondary hypogammaglobulinemia (SHG), and recurrent bacterial infections should be considered for immunoglobulin replacement. These recommendations are based on several observations. Firstly, the most common complication and cause of death in CLL is infection, which often occurs in patients with advanced disease and hypogammaglobulinemia. Secondly, hypogammaglobulinemia is prevalent in CLL; in one study, at least one isotype (IgG, IgM, or IgA) was found to be abnormally low in 48/50 patients (96%) [48]. Thirdly, earlier studies have indicated that patients with CLL who were at increased risk of infections, when treated with immunoglobulins, had significantly fewer bacterial infections than patients who received a placebo. This study also showed that the period from study entry to the first severe bacterial infection was longer in patients who received immunoglobulins [49].

The American College of Allergy, Asthma & Immunology (ACAAI) recommends that patients with CLL with recurrent serious bacterial infections who are hypogammaglobulinemic with suboptimal protective antibody levels following immunization to diphtheria, tetanus, or pneumococcal infection should be considered eligible for immunoglobulin replacement therapy [37]. These recommendations are in line with NCCN guidelines for CLL where IVIG replacement therapy is recommended for selected patients with serum IG < 500 mg/dl and recurrent sinopulmonary infections requiring intravenous antibiotics or hospitalization [34,45].

The standard IVIG dosage for infection prevention in CLL patients is 400 mg/kg administered every three to four weeks. The monitoring of IgG trough levels is recommended to ensure adequate immune protection, with levels maintained at or above 500 mg/dL to optimize infection risk reduction [37]. Subcutaneous immunoglobulin (SCIG) is a newer valid treatment option that has been studied extensively in patients with Primary Immunodeficiencies. SCIG allows self-administration at home, and it proved to be as well tolerated as and non-inferior to IVIG in patients with CLL in a study by Visentin et al. [50].

In addition to reducing the risk of recurrent infections, IVIG may also play a role in mitigating pulmonary complications in select patients. Bronchiectasis is an underrecognized complication in CLL, often linked to hypogammaglobulinemia and recurrent infections. A recent review reported a prevalence of 8–10%, with symptoms including chronic cough and recurrent pneumonia [51]. High-resolution CT should be considered in symptomatic patients. IVIG may reduce the infection burden in those with bronchiectasis and should be part of a multidisciplinary approach, including pulmonology referral and airway clearance strategies.

### 5.3. Lifestyle Interventions

Tobacco cessation is an essential component of supportive care for patients with CLL, as smoking can exacerbate immune dysfunction and increase the susceptibility to infections and secondary malignancies. NCCN recommends that all patients with cancer who smoke—regardless of their readiness to quit—be offered a comprehensive cessation treatment. This includes evidence-based behavioral counseling, pharmacotherapy (e.g., combination Nicotine Replacement Therapy or varenicline), and close follow-up. While some may use non-combustible nicotine products like e-cigarettes to reduce harm, they are not FDA-approved for cessation, and the goal remains complete abstinence from smoking tobacco. Tobacco cessation should be integrated across the cancer care continuum, with documentation of smoking status and relapse support. NCCN emphasizes that quitting is never too late, as cessation improves treatment outcomes and reduces recurrence and second malignancies [52].

Nutrition also plays a crucial role in supporting overall health in patients with CLL. Growing evidence from breast cancer survivorship research suggests that overall dietary patterns, rather than individual nutrients, play a critical role in long-term outcomes. Healthful dietary patterns—such as those consistent with the Healthy Eating Index or Mediterranean-style diets—are associated with reduced all-cause and non-cancer mortality, while Western dietary patterns (high in red/processed meats, refined grains, and added sugars) are linked to worse outcomes. Although most data come from solid tumor populations, these findings underscore the broader relevance of plant-forward, nutrient-dense diets for cancer survivors, including those with CLL. Such diets may help mitigate treatment-related comorbidities, support immune function, and improve overall health—a critical consideration given the high burden of frailty and immune dysfunction in CLL. Importantly, postdiagnosis dietary improvements offer survival benefits, reinforcing that it is never too late to adopt healthier eating behaviors in cancer survivorship. The ACS guidelines emphasize the importance of a dietary pattern rich in vegetables, fruits, and whole grains while minimizing saturated fats and ensuring adequate fiber intake [53].

Physical activity has been shown to improve quality of life and reduce fatigue in patients with CLL. At our center, the HEALTH4CLL program—a structured lifestyle intervention for patients with CLL—has demonstrated significant benefits in improving fatigue, physical function, and overall well-being. The program incorporates both aerobic and resistance training, aligned with ACS guidelines recommending at least 150 min of moderate-intensity exercise and strength training twice weekly. It also includes a dietary component adapted from the Diabetes Prevention Program, focusing on a reduced fat and calorie intake and an increased consumption of fruits, vegetables, whole grains, and lean proteins, and limiting processed foods and added sugars. Participants are encouraged to use self-monitoring to track food intake and weight. Among 31 participants, the most effective outcomes were seen with the combination of exercise and daily self-monitoring, supporting the feasibility and impact of low-intensity, remote lifestyle interventions in CLL survivors [54].

## 6. Future Directions

As patients with CLL live longer, there is a growing need for cancer screening strategies that reflect their unique clinical challenges—particularly their heightened risk of other primary malignancies and compromised immune function. Future studies should assess whether more frequent screening for skin, lung, and gastrointestinal cancers could lead to earlier detection and better outcomes in this population. In parallel, it is important to evaluate how emerging cancer screening tools perform in immunocompromised patients like those with CLL.

While current guidelines and expert consensus offer valuable direction for preventive care in CLL, several gaps are seen. First, there is a lack of robust, prospective data evaluating the long-term effectiveness of cancer screening, vaccinations, and IVIG replacement in patients with CLL and also in patients with CLL receiving targeted therapies such as BTK or BCL-2 inhibitors. Most data are derived from small observational studies or broader oncology guidelines that are not specific for patients with CLL. Second, few studies have evaluated whether preventive cancer screening strategies improve overall survival or long-term quality of life outcomes in this population. Third, many existing studies include heterogeneous patient populations and lack standardized criteria for inclusion, making it difficult to generalize findings across treatment eras or geographic regions. These limitations underscore the need for prospective trials and real-world studies that focus specifically on patients with CLL and also include patients treated with targeted agents—to allow the development of evidence-based preventive care recommendations for patients with CLL.

Vaccine optimization remains another key area. Given the suboptimal immune responses seen in many patients with CLL, especially those receiving targeted therapy, future research should investigate the role of vaccine adjuvants, booster regimens, and novel immunization strategies. Finally, as targeted agents such as BTK and BCL-2 inhibitors continue to reshape treatment, we need a better understanding of how they influence infection risk and immune recovery over time. Carefully tailored supportive care—including antimicrobial prophylaxis, immunoglobulin replacement, and personalized vaccine schedules—will be essential to improving outcomes and quality of life.

## 7. Conclusions

With advances in therapy, patients with CLL are living longer, necessitating the development of a comprehensive survivorship care. Immune dysfunction remains a key challenge, increasing the susceptibility to infections and malignancies. Infection prevention through vaccination, antimicrobial prophylaxis, and IVIG therapy can play a critical role in reducing morbidity. Evidence-based cancer screening can also play a crucial role in mitigating morbidity and mortality among patients with CLL. Personalized screening approaches should be integrated into long-term care plans, considering the individual risk factors, treatment history, and overall health status. Healthcare providers must remain vigilant in implementing preventive strategies, including routine screening and patient education, to optimize outcomes in this high-risk population.

## Figures and Tables

**Table 2 cancers-17-02093-t002:** Suggested vaccine recommendations and special considerations for patients with CLL.

Vaccine	Timing	Schedule	Special Considerations in CLL	References
Pneumococcal	Before treatment or during disease control	PCV-20 single dose or PCV-15 + PPSV23 (8-week interval) and repeat PPSV23 5 years after the first dose and one final dose after 65 years	Recommended at any age.	CDC Schedule
Influenza	Annually, preferably before flu season	Single annual dose	High dose inactivated vaccine is recommended.	CDC Schedule [28],
COVID-19	Before treatment; booster doses as per guidelines	Primary series (2–3 doses) + booster doses as per guidelines	High-risk patients may need additional doses or monoclonal antibody therapy.	CDC Schedule [28],
Herpes Zoster	Before treatment or during disease control	2 doses, 2–6 months apart (or 4 weeks if rapid protection needed), in adults older than 50 years	Consider at any age; live zoster vaccine should be avoided.	[28]
RSV	Before treatment or during disease control	All adults aged 75 and older	Adults aged 60–74.	CDC Schedule

## Data Availability

No new data were created or analyzed in this study. Data sharing is not applicable to this article.

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
