# Peer review of "Best Supportive Care for Patients with Chronic Lymphocytic Leukemia: Relevance of Cancer Screening and Immunizations"

_cancers, 2025, doi:10.3390/cancers17132093_

Round 1

Reviewer 1 Report

Comments and Suggestions for Authors

This article provides an overview for clinicians on preventative measures for infection and secondary malignancy in haematology patients. It would be a useful guidance and condenses in one place which I believe is a good thing and currently lacking.  

- Overall, the flow of the document works well but the section on immune dysfunction in CLL is a bit disorganised and could be rearranged to help the reader. For example we start off in first para describing hallmarks of CLL immune deficiency, then jump to treatment impacts, then back on line 73 discussing more general CLL immune dysfunction. It would be far better in my opinion to introduce with the first para, then go into the general info on CLL from line 73, then discuss treatment impact or even subheading these areas? The vast majority of CLL patients are W+W where the latter paragraphs of that section is far more relevant. If this guideline is considered for American practice only then I believe it should be stated so OR an appreciation given that services / vaccine availability based on licenses and reimbursement methods will differ between countries needs to be mentioned.  

  • I have some concerns about the lack of primary reference data. The guidance provided isn't always based on the licensing and the indications for the vaccines will differ by country e.g RSV vaccination is listed as >60 but the license for is >50 for adjuvated RSV. Similarly for the section on screening - no references in the table which is what many will refer to- a 6-12 month review for all CLL patients with dermatology for example. There doesn't seem to be an appreciation of how practice is determined by policy and the guideline produced is not going to be easily followed by all. Would it not be better to have the reference/ license indications for each drug vaccine they are proposing?
  • Similarly, low dose CT scan for high risk smokers... referenced by a guideline. Is there any data for lower income countries that CXR would also be beneficial for this group? CT/CXR relative risk from ionising radiation would be interesting to add vs difference in detection rates. 
  • No mention about bronchiectasis risk or additional management 
  • Hepatitis B vaccination is recommended to all in this guideline but this is not the case. Hep B immunisation is usually based on the individual country's prevalence for Hep B and individual risk assessment of the likelihood of coming into contact with it. 
  • Data around the benefit of booster vaccine with prevnar in CLL - may be available to some in a private setting and from what we have learnt from COVID vaccination one would assume would be more beneficial. 
  • The data on BTKi and HepB Vs VZV is a bit misrepresented- the point is that a memory recall response is likely to be more preserved on BTKi (VZV) than a denovo reponse that requires naive B cells (Hep B). This is important to point out to readership as it may infer  long lived plasma cells to measles etc will provide protection for other childhood infections for those on BTKi and that boosting responses is  important.  https://doi.org/10.1182/blood.2020008758
  • Hep B prophylaxis to anyone starting B cell depleting treatment who has positive serology should be clearly stated. 
  • Similarly herd immunity is such that HiB is very rare now and some countries have dropped this vaccine. Other non-vaccine preventable haemophilus are more common for pneumonia in these patients
  • There are several papers on BTKi and risk of aspergillus/PJP. Initial reports were in heavily treated and Ibrutinib where there are much more off target effects. Subsequent data has shown the risk is very small in non-ibrut patients. Mindful of this, emphasising PJP association with BTKi I think isn't helpful and needs to be more balanced.  

    https://doi.org/10.1182/bloodadvances.2020001678

    10.1182/blood-2017-11-818286. 3 https://doi.org/10.1080/10428194.2022.2056179

  • A section on timing of vaccination focuses on at diagnosis/ early and on continuous BTKi. What about those who have had treatment? 
  • No mention of the recent IMPROVE trial that has shown pausing BTKI doesn't improve responses either- a practice that has crept in from observational COVID  data (https://doi.org/10.1016/S2352-3026(25)00008-0)

Reviewer 2 Report

Comments and Suggestions for Authors

The review “Best supportive care for patients with Chronic Lymphocytic Leukemia: relevance of cancer screening and immunizations” proposed by Vanthana Bharathi and Alessandra Ferrajoli offered an overview regarding the management of CLL patients in the context of prevention and support.

I propose to improve this review with some additional information and according to several observations.

No recent or unpublished data on the effectiveness of prevention strategies (screening, vaccines, IVIG) specifically in CLL patients treated with the new targeted therapies are presented. Coud authors add this information?

The long-term effects of the proposed prevention strategies are not discussed in detail, nor their impact on overall survival or quality of life.

A final section on methodological limitations and data gaps would help to better contextualize the recommendations.

The inclusion/exclusion criteria of the cited studies are missing.

It is not specified if a systematic or narrative search was conducted. A clearer methodology could be written, such as the databases consulted and the period considered; any search strategies (e.g. keywords, filters applied).

I suggest preparing decisional flowchart or summary tables to guide the clinician in choosing vaccinations or screening based on the patient's status.

Reviewer 3 Report

Comments and Suggestions for Authors

• What is the main question addressed by the research?

- This is a concise summary on cancer screening strategy and vaccination recommendations for patients with chronic lymphocytic leukemia. 

• Do you consider the topic original or relevant to the field? - Yes
Does it address a specific gap in the field? - Yes

• What does it add to the subject area compared with other published material? 

- An update on preventive measures for CLL patients such as  oncoscreening and immunization is a desirable addition to the literature.

• What specific improvements should the authors consider regarding the methodology?
- No specific improvement of the methodology is needed.

• Are the conclusions consistent with the evidence and arguments presented and do they address the main question posed? Please also explain why this is/is not the case.
Yes

• Are the references appropriate?
Yes

• Any additional comments on the tables and figures.

- No figures are provided, the display items consist of the two tables. It would be desirable to add also to the table the references supporting the corresponding recommendations on either cancer screening of vaccination. 

Author Response

No figures are provided, the display items consist of the two tables. It would be desirable to add also to the table the references supporting the corresponding recommendations on either cancer screening of vaccination. 

Thank you for your comment. The references have been added to the table.